# Stability Performance Analysis of Various Packaging Materials and Coating Strategies for Chronic Neural Implants under Accelerated, Reactive Aging Tests

**DOI:** 10.3390/mi11090810

**Published:** 2020-08-26

**Authors:** Yan Gong, Wentai Liu, Runyu Wang, Matthew Harris Brauer, Kristine Zheng, Wen Li

**Affiliations:** 1Department of Electrical and Computer Engineering, Michigan State University, East Lasing, MI 48824, USA; liuwentai0629@gmail.com (W.L.); wangruny@msu.edu (R.W.); brauerma@msu.edu (M.H.B.); wenli@msu.edu (W.L.); 2Massachusetts Institute of Technology, Cambridge, MA 02139, USA; zhengkristine@gmail.com

**Keywords:** accelerated reactive aging test, long term, stability, packaging, implants, Parylene C, SiO_2_, Si_3_N_4_

## Abstract

Reliable packaging for implantable neural prosthetic devices in body fluids is a long-standing challenge for devices’ chronic applications. This work studied the stability of Parylene C (PA), SiO_2_, and Si_3_N_4_ packages and coating strategies on tungsten wires using accelerated, reactive aging tests in three solutions: pH 7.4 phosphate-buffered saline (PBS), PBS + 30 mM H_2_O_2_, and PBS + 150 mM H_2_O_2_. Different combinations of coating thicknesses and deposition methods were studied at various testing temperatures. Analysis of the preliminary data shows that the pinholes/defects, cracks, and interface delamination are the main attributes of metal erosion and degradation in reactive aging solutions. Failure at the interface of package and metal is the dominating factor in the wire samples with open tips.

## 1. Introduction

Implantable microelectronic devices have been widely used in neuroscience and clinical research for manipulating and mapping of neural activities [1,2]. Many implants target a long-term period of > 20 years in the host body, which, however, has not yet been achieved due to the instability of the device’s package [3]. Assessment of long-term device stability provides a better understanding of the underlying factors that cause implant failure or complications [4], but requires a long testing cycle. As an alternative, reactive accelerated aging (RAA) testing in hydrogen peroxide (H_2_O_2_) solutions can effectively shorten the testing cycle while mimicking acute post-surgery inflammatory reactions [5,6]. Ideally, a packaging material should fulfill the following requirements: electrically insulative to prevent cross-talking, biocompatible, chemically resistant, low moisture, and gas permeation [7,8,9,10,11,12,13]. Among various materials, Parylene-C (PA) is one of the most prevailing packaging materials for biomedical implants [14,15,16,17,18]. SiO_2_ [19,20,21,22], and Si_3_N_4_ [23,24,25] are usually used as insulating layers in silicon based neural probes as well as an encapsulation protective layer to protect microelectronics from corrosive environments such as body fluids. However, the packaging performances of these films have not been studied systematically. In the packaging area, although various sealing methods [26] and materials [27] have been studied and discussed for many years, the probability of device failure should continue to increase over time, and the failure of the device package is a gradual process. There is still room for research on this gradual process. Therefore, an in-depth understanding of the performance and stability of these insulating materials is meaningful for the future development of chronic implants.

In this work, we investigated the reliability performance of three materials and their combinations in three controlled environments. We analyzed the performance during the RAA test by cumulative failure probability and scanning electron microscopy (SEM) images. Tungsten wires were coated with single or multiple layers of PA using chemical vapor deposition (CVD) by Specialty Coating Systems™ PDS 2010 Parylene Coating System (Specialty Coating Systems, Inc., Indianapolis, IN, USA), as well as SiO_2_ and Si_3_N_4_ using plasma-enhanced chemical vapor deposition (PECVD) (Plasmalab 80 Plus™, Oxford Instruments Plasma Technology, Oxford, UK). Two device configurations were investigated: closed- and open-tip wires. PA coatings with different thicknesses were tested. During the experiments, the samples were aged in phosphate-buffered saline (PBS), PBS + 30 mM H_2_O_2_ and PBS + 150 mM H_2_O_2_ [28,29] at ~67 °C [30]. Similarly, a room temperature (22 °C) comparison group with a closed-tip 3 µm PA coating wires was also added to the experiment. Furthermore, inorganic coating (SiO_2_, Si_3_N_4_) and their composite strategies were also tested. The experiment used the same design and solutions (PBS, PBS + 30 mM H_2_O_2_, and PBS + 150 mM H_2_O_2_) as PA coating to compare the performance difference of inorganic materials encapsulation and PA, and investigate the potential of the composite strategy. Electrochemical impedance spectroscopy (EIS) was used to monitor impedance changes which characterize the integrity of packages. The mean-time-to-failure (MTTF) was determined when the measured 1 kHz impedance of the samples changed over 50% of the initial value. All data were processed by Weibull parameter estimates (wblfit, MATLAB) to calculate the parameters of the Weibull distribution, and then fitted using polynomial or Fourier fitting to extrapolate discrete Weibull cumulative probability data. The performance of various packaging strategies was represented by the failure probability of the sample over time. Optical microscope and scanning electron microscopy (SEM) were used to identify physical damage of the coatings at the time point of wire failure. More details are given in the following sections. The main goal of this research is to build a test framework, by combining the RAA test method and a Weibull probability model to evaluate the risk and progression of device failure. In the future, we will apply this framework to evaluate other organic and inorganic materials that are widely used in biomedical device packaging, such as polyimide [31,32,33,34,35,36], SiC [37,38], diamond-like carbon [39,40], etc.

## 2. Methods

### 2.1. Chemicals

PBS (1 M, pH = 7.4) was purchased from Sigma-Aldrich (St Louis, MO, USA). Concentrated H_2_O_2_ (30%) was purchased from Chemistry Stores at Michigan State University (MSU), both used as received. Solutions were diluted by deionized (DI) water (16 MΩ·cm).

### 2.2. Testing Probes

Testing wires were made of common neural implant probe materials (tungsten) to simulate the corrosion probes during in vivo testing to evaluate the performance of each package method. Due to the strength, rigidity, and recording capability of tungsten, 50 µm diameter microwire becomes an excellent choice for intracortical applications [41]. Tungsten wire in this study was used as an electrode target to simulate the internal metal structure. High purity tungsten wire (99.95%) was used, conforms to ASTM F288-96, Type 1A. The wire is 5.5 cm in length and 61 µm in diameter. The microwires were coated with single or multiple layers of PA, SiO_2_, and Si_3_N_4_. SiO_2_ was deposited in a 13.56 MHz driven parallel plate reactor PECVD system Plasmalab 80 Plus with manual sample loading and a heated substrate electrode using the following conditions: process gases of SiH_4_ with flow rate of 170 sccm, N_2_ with flow rate of 170 sccm, and N_2_O with flow rate of 710 sccm, working pressure of 1000 mTorr, substrate temperature of 300 °C, high-frequency RF power of 20 Watts with a deposition rate of 8.3 Å/s. Si_3_N_4_ was deposited in the same PECVD reactor using the following conditions: process gases of SiH_4_ with flow rate of 400 sccm, N_2_ with flow rate of 400 sccm, and NH_3_ with flow rate of 20 sccm, working pressure of 1000 mTorr, substrate temperature of 300 °C, high-frequency RF power of 20 Watts with a deposition rate of 4.1 Å/s. PA was deposited in a Parylene deposition system (Specialty Coating Ststems™ PDS 2010 Parylene Coating System) under the following conditions: base pressure of 14 mTorr, deposition pressure of 22 mTorr, furnace temperature of 690 °C, and vaporizer temperature of 175 °C. Two configures were investigated: closed-tip and open-tip wires. For the open-tip samples, the insulation layer at the tip of the wire was precisely stripped to mimic the electrode structure. To fabricate the open-tip samples, the tip of the tungsten wires (3 mm) was clamped between multiple glass slices in the PECVD chamber. Under this setting, the glass slices physically prevent the deposition of the insulating material, and reduced additional damage that may be caused by manual stripping the insulation layer. The fully enclosed (closed-tip) type is aimed to fully understand the performance of various packaging materials and different packaging strategies. Several closed-tip design groups were applied, in which the wire tip was completely encapsulated without any artificially induced opening.

To design a comprehensive experiment for packaging materials and strategies studying, the following three categories based on different deposition procedures were tested for a total of 14 test groups (Figure 1). First, a single deposition design is planned to simply test (no pretreatment on the microwires) the performance of different packaging materials under the same conditions. This design contains only one material (SiO_2_, Si_3_N_4_, or PA) as a wire package. For the PA group, experiments were also performed for variable thickness (1, 3, 5 µm). A total of 90 electrodes were tested in three different solutions for three different materials, with 10 samples tested under each condition. Second, a composite deposition design is used to study whether the superposition of two materials can effectively improve the survival rate of open-tip samples. Previous studies show that composite deposition of multilayer films can significantly elongate the percolative pathway compared to their homogenous layer counterparts [12]. The tight ceramic layer allows diffusion only through its pinholes and cracks. Thereby, the corrosive solution that penetrates the protective layer is significantly reduced in both quantity and rate. This design involves two material combinations (SiO_2_ + PA or Si_3_N_4_ + PA) in one thickness (100 nm + 1 µm). A total of 60 electrodes were tested in three different solutions for two different combinations. Last, multiple stack deposition using a single polymer material is designed and tested. We hypothesize that the second polymer coating can repair the defects from the first layer and disrupt the continuity of the pinholes, therefore improving the packaging performance. In this work, stacked deposition samples superimposed three stacks of 1 µm PA was prepared. A total of 72 electrodes were tested in three different solutions. The details of all test samples and their configurations are shown in Table 1.

### 2.3. Dissolution of Tungsten by Hydrogen Peroxide

Elemental tungsten has very stable characteristics and is chemically resistant to corrosion from oxygen, acids, and alkalis [42]. However, when immersed in the H_2_O_2_ solution, tungsten will generate a dissolution effect [43], and eventually, actual erosion phenomenon occurs. This unique dissolution phenomenon is another major reason for choosing tungsten wires. The dissolution of tungsten in H_2_O_2_ can be expressed by the following reaction Formulas (1)–(3) [44]:W(s) + 2H_2_O_2_ → WO_2_(s) + 2H_2_O,(1)
2WO_2_(s) + 6H_2_O_2_ → H_2_W_2_O_11_(aq) + 5H_2_O,(2)
3H_2_W_2_O_11_(aq) + 7H_2_O → 2H_2_W_3_O_12_(aq) + 8H_2_O_2_,(3)

The erosion rate is directly proportional to the reaction temperature in a certain range, and approximately 60 °C was found as the most rapid dissolution temperature [43]. This is close to the reaction temperature set in our experiments (67 °C). Therefore, the open-tip design that directly exposes the tungsten wire will face the problem that the reaction may be too intense such that the encapsulated tungsten could be rapidly corroded. The rapid erosion rate makes the test results change too fast to be measured for accurate comparisons of the failure trends of various packaging materials and strategies. This is also one of the necessities of the closed-tip design.

### 2.4. Experiment Setup

A modified Pyrex 1-cup (237 mL) square glass vessel and coordinating Snapware lids (Item #: 1109305, Snapware Corp, Mira Loma, CA, USA) was used as a reaction chamber. Six glass nozzles were installed on the top lid as loading ports. The lid was sealed with epoxy, and PDMS was poured on the lid to prevent vapor leakage. The reaction chamber was connected to two dosing pumps (INTLLAB, Shenzhen, Guangdong, China) to precisely control the inlet and outlet flow rates of the reaction chamber. The glass chamber was placed on a hot plate (Thermo Scientific^TM^, Waltham, MA, USA). For thermal aging, the thermostat temperature of the hotplate was set to a temperature higher (95 °C) than the designed temperature (67 °C) to compensate for convective heat loss, and the solution was stirred at 75 rpm using a magnetic stirrer to ensure uniform mixing. The glass container was covered with aluminum foil to eliminate the effect of light on the reaction. The tungsten wires were fixed on the robber holder with the tips (~30 mm) immersed into the reaction media. The experimental setup schematic diagram is shown in Figure 2. During the RAA test, the samples were aged in PBS, PBS + 30 mM H_2_O_2_ and PBS + 150 mM H_2_O_2_. Different concentrations of hydrogen peroxide were utilized to simulate mild or severe inflammation conditions, because the immune system can create a very aggressive chemical environment in the brain, rich in digestive enzymes and reactive oxygen species (ROS) [45,46,47]. The chosen temperature was below the glass transition temperature of PA to prevent thermally induced PA degradation [48]. According to Arrhenius modeling of reaction acceleration [30], a simulation physiological aging at an 8x acceleration factor *f* can be calculated using Equation (4):*f* = 2^Δ*T*/10^, Δ*T* = *T* − *T_ref_*,(4)
where *T* is the testing temperature, *T_ref_* is a reference temperature, and *f* is the acceleration factor.

To better understand the role of temperature in the aging test on the packaging performance, a room temperature (22 °C) comparison group was also added to the experiment using closed-tip 3 μm PA coated tungsten wires. The testing wires were sealed in a flat bottom headspace vial (ALWSCl Technologies, Zhejiang, China) which was filled with different reactive solutions (PBS or PBS + H_2_O_2_) under the same concentrations as used in the high-temperature groups. Vials were stored in a dark box to eliminate light interference. The solution concentration and the sample condition were checked daily.

### 2.5. Hydrogen Peroxide Concentration Maintaining

Under the existing conditions, the kinetics of H_2_O_2_ degradation at preset temperature (67 °C) follows first-order kinetics with a half-life of 143 min. In our flow cell design (Figure 2A,B), the dosing pump constantly injected high concentration H_2_O_2_ + PBS solution into the reaction chamber to keep constant H_2_O_2_ concentrations. The solutions in the reaction chamber and the storage bottles were replaced every three days to ensure a stable concentration. At the beginning, the reaction chamber was filled with 200 mL of the designed concentration of H_2_O_2_ + PBS solution (PBS +30 mM H_2_O_2_ or PBS + 150 mM H_2_O_2_). During the experiment, the high concentration solution is continuously pumped into the reaction chamber. In particular, the PBS + 90 mM H_2_O_2_ solution was pumped into the reaction chamber with an inflow rate of 38 mL/min and an outflow rate of 32 mL/min to maintain the concentration of the PBS + 30 mM H_2_O_2_ solution, while the PBS + 375 mM H_2_O_2_ solution was used for the PBS + 150 mM H_2_O_2_ solution with 40 mL/min inflow and 38 mL/min outflow rates. Four pumps were controlled by smart switches (KMC, Hangzhou Kaite Electrical Appliance Co., Ltd., Hangzhou, China) and operated in 30 s/60 min ON/OFF duty cycle to circulate the solution.

The testing solutions were sampled twice a day to analyze the concentration of H_2_O_2_ in the reaction chamber using a titanium oxalate assay [49]. The basic operation procedure was to mix 50 μL sample solution with 200 µL, 1 M H_2_SO_4_, and 200 μL titanium potassium oxalate (50 g·L^−2^), which was then diluted to 5 mL with deionized water. After waiting for 5 min until the reaction is complete, ultraviolet-visible spectroscopy (UV-Vis) were taken and adsorption at 390 nm was measured by a SpectraMax^®^ M3 Multi-Mode Microplate Reader (Molecular Devices LLC, San Jose, CA, USA). As shown in Figure 2C, a constant concentration of ~30 mM can be maintained using the flow cell, at the same level as the standard solution in absorbance over 3 days. The 150 mM testing solution was evaluated using the same protocol and its concentration curve is similar to that of the 30 mM solution (data not included). Considering that the pH value will also affect the package survival rate, the pH values of the three solutions were tested (Hydrion™ Insta-Chek™ pH Test Paper 0.0 to 13.0, Micro Essential Lab, Brooklyn, NY, USA): ~7.4 for PBS, ~7.2 for PBS + 30 mM H_2_O_2_ solution, ~7 for PBS + 150 mM H_2_O_2_ solution. It can be seen that the concentration of H_2_O_2_ has little effect on the overall pH value.

### 2.6. EIS Measurement

EIS was performed on an Autolab PGSTAT128N potentiostat (Metrohm Autolab, Herisau, Switzerland). All measurements were done in PBS (0.01 M, pH ~ 7.4) at room temperature. Impedance data were measured using a three-electrode configuration, where an Ag/AgCl wire was used as a reference electrode, a Pt wire was used as a counter electrode, and the tested tungsten wire was used as a working electrode. A sinusoidal waveform with amplitude of 10 mV (root mean square) was applied in a frequency range of 1 Hz to 100 kHz. Figure 3 gives an example of EIS measurements, showing the impedance changes of the 1 µm PA packaged open-tip tungsten wire on day 0 and day 7 in PBS + 30 mM H_2_O_2_ solution. It can be seen that the wire impedance at the high frequency of > 100 Hz increased significantly by orders of magnitude after 7 days, mainly due to the dissolution effect and discontinuous structures of the wire. The significant change in phase was from a sinusoidal-like curve to a relatively flat curve, and the overall phase values were increased.

### 2.7. SEM Image

At the endpoint of each testing, tungsten microwires were carefully removed from the reactive chamber, washed with DI water to remove the residual from reaction and saline, and inspected using a Hitachi S-4700II Field Emission SEM (Hitachi High Technologies America, Schaumburg, IL, USA) with an electron acceleration potential of 15 kV. This FE-SEM provides high magnification and high precision (1.5 nm) microscopic imaging capability to visualize the erosion of the tungsten implants.

### 2.8. Data Analysis

Defining failure points for impedance variation is the first step in data analysis. Having considered different damage tolerances of different devices, this article uses a relatively low failure threshold to verify the maximum protection time of different packaging materials and strategies. The MTTF was determined when the measured 1 kHz impedance of the samples changed over 50% of the initial value. Furthermore, the data were processed by Weibull parameter estimates (wblfit, MATLAB) to calculate the parameters of the Weibull distribution. The Weibull distribution gives a distribution for which the failure rate is proportional to a power of time [50,51,52]. Then polynomial fitting or Fourier fitting was used to fit discrete Weibull cumulative probability data. Results were represented by the failure probability of the sample over time. The cumulative probability curve shows the overall risk of the sample failing during the experimental time. When the first sample fails, it means that, under the same design and same test condition, the sample has a probability of failure, and the probability is determined by the ratio of the number of failures at that time to the total number of experiments. For example, in Figure 6A, the failure probability on the seventh day of the Si_3_N_4_ coated samples is higher than 80%. This means that this packaging strategy has a high risk of failure although the sample may not necessarily fail within 9 days. It is of note that the probabilities curve is lower than 0 only because of the fitted curve model. The fitting curve is added to the chart to better indicated the trend of the sample failures. The actual point of the data will not be lower than 0.

## 3. Results

### 3.1. Temperature Effect

Temperature is one of the key parameters of the accelerated aging test. To verify the effects of temperature on subsequent experiments, a temperature comparison test was designed and implemented. Closed tip tungsten microwires coated with 3 µm PA were aged at 67 °C and 22 °C in three different solutions for one week. The 3 µm coating was designed to prevent the tungsten wire from H_2_O_2_ erosion too fiercely. A thicker protective coating can slow down the erosion rate to a certain extent, resulting in a smooth impedance change from the measurement. Figure 4 shows the stability performance of the tungsten wires tested at two different temperatures.

In Figure 4A, it can be seen that the room temperature samples soaked in 1× PBS still maintained about 60% survival probability (100%-failure probability) after 7-day soaking. This failure probability did not change after the fourth day, indicating that, after the first four day’s stability fluctuation, no new failed samples were detected. The failure rate of the room-temperature group in 3 different solutions was significantly lower than that of the high-temperature group for a certain period of time. It is worth noting that, as shown in Figure 5A,B, the room-temperature group began to occur failure incident after a certain day’s delay, thus reflecting the accelerated effect of temperature in the experiment. According to the high-temperature group data, as shown in Figure 4A, unlike the room temperature group which failure rate stabilized in a short period (day 4), the failure rate of the high-temperature group continued to increase even after day 7, and is expected to reach a stabilized failure rate much higher than that of the room temperature group. Similar results were obtained from the samples tested in H_2_O_2_ + PBS solutions, as shown in Figure 4B,C. While preliminary, these results suggest that high temperature indeed accelerates the aging test, and therefore, the failure of the PA package.

### 3.2. Analysis of Three Single-Layer Materials at 1 µm Thickness

For the performance study of different packaging materials, one organic material (PA) and two non-organic materials (Si_3_N_4_ and SiO_2_) were selected intentionally. The same package thickness (1 µm) was applied, and the same experimental environments were maintained for those three materials. After seven days of experiments, the cumulative failure probability results are shown in Figure 5. Overall, because of the fast dissolution effect in the open-tip design as mentioned previously, the failure rates of the open-tip tungsten wires were very high (>90%) under mild and severe inflammatory conditions containing H_2_O_2_, as shown in Figure 5B,C. Most of the tested sample failed on the first day of soaking, and all the samples were damaged on Day 2.

Figure 6 shows the changes in average 1 kHz impedance as an indicator of the stability of the tungsten wire under the protection of different materials. In general, the impedance change is related to the exposed area of the tungsten wire, and the larger the exposed area, the more impedance decreases. However, due to the dissolving effect of H_2_O_2_ on tungsten, the exposed wire could be completely dissolved or broken, resulting in increased impedance at the later stage of testing. Therefore, both abnormal impedance decrease and increase were considered as sample failure, as shown in Figure 6. Combining the information from Figure 5A and Figure 6A, it can be concluded that, in the PBS solution, the stability of the tungsten wires protected by PA is excellent. All the PA-coated samples remained intact within seven days, and their impedance changes were considered minimal. When the solution cannot rapidly dissolve tungsten and PA adheres tightly surrounding the wire, PA can effectively slow down the expansion of the exposed area on the tip and prevent the wire from being eroded. Despite the large scale failure of the other two packaging materials (Si_3_N_4_ and SiO_2_), as shown in Figure 6A, the impedance change returned to a stable state after the third day. One possible reason is that, although the protective layer does not protect the tungsten wire effectively in the first time, it can still control the destruction area within a certain range, thereby keeping the impedance stable to a certain extent.

In a simulated inflammatory environment (H_2_O_2_ + PBS) where the solution can quickly attack the tungsten material, the situation is completely different. As shown in Figure 6B,C, none of the three protective materials can protect the wire samples for more than seven days. In 30 mM H_2_O_2_ + PBS, the PA protective layer shows slightly better packaging performance than SiO_2_ and Si_3_N_4_. PA can provide effective protection for 24 h with a failure rate of < 30%. In 150 mM H_2_O_2_ + PBS, a large number of failure incidents occurred for all three packaging materials on day 1, and the impedance changes of the wires were dramatic. For the impedance fluctuation in Figure 6C, the initial impedance drop is the result of the increased exposed metal area, due to package damage or metal corrosion along the bonding interface. The subsequent increase in impedance occurs when the wire becomes thinner or completely broken due to metal erosion, resulting in a decreased metal area. It is also worth noting that a large number of metal fractures were also observed in the tungsten wires due to reaction with H_2_O_2_. The SiO_2_ coated samples were completely broken on the sixth day under the mild inflammatory environment (30 mM H_2_O_2)_, while all the samples failed in six days under the severe inflammatory environment (150 mM H_2_O_2_).

Figure 7 shows a comparison of the impedance change and wire breakage among the wire samples protected with various packaging materials. It can be seen that an abnormal impedance increase occurred right before wire breakage. For example, on day 5 before the massive fracture of the PA coated sample, their impedance increased abnormally by over 3–10 times. This phenomenon can be explained that, before the metal fracture occurs, many tiny cracks have already occurred inside the wire and caused discontinuous structures that greatly increase the impedance.

Compared with the open-tip design (Figure 5), the closed-tip design shows its unique failure rate curve. The function of the closed-tip probe is to reduce the dissolution effect of H_2_O_2_ on the tungsten wire, thereby slowing down change in the probability fitting curve. By slowing down the change curve of failure probability, the cumulative probability curve of each material does not overlay within a certain time, so that the protection performance of each material can be compared in detail. It must be pointed out that the closed-tip design reduces the dissolution effect of H_2_O_2_ but does not completely prevent the aging effect. The package will still age in the RAA test and fail at a certain point, resulting in the impedance changes of the wire.

As shown in Figure 8A, in PBS, the PA coated, closed-tip wires show poor stability than the open-tip counterparts (Figure 5A). Since the closed-tip sample is insulated, its initial impedance is very high (>1 × 10^7^ Ω). Once the solution contacted tungsten through the damage site, the sample changes from an insulator to a conductor, resulting in dramatic impedance drop. The failure threshold is based on the impedance changes, so the closed-tip is relatively more sensitive to package damage in the PBS solution. For the open-tip design, since it has an opening area, the impedance changes mainly due to the change in the exposed area when the damages to the tungsten wire occurred, which is relatively slow in the PBS solution. Therefore, the impedance change of the closed-tip samples is not as severe as the closed-tip ones. However, when the open-tip samples are tested in the H_2_O environments (30 mM H_2_O_2_ + PBS, and 150 mM H_2_O_2_ + PBS), direct exposure of the tungsten wire to H_2_O_2_ can cause severe damages in the exposure area, therefore resulting in significant increase or decrease of wire impedance. Although the closed-tip sample also had failure incidents, it still shows relatively better stability than the open-tip sample.

In general, the closed-tip PA protective layer can protect the device effectively in PBS within 7 days with a failure rate of < 40%, and the probability of failure climbed in the first 3 days and stabilized below 40% on day 4 (Figure 8A). Under mild inflammation environment (Figure 8B), the failure probability of PA packaging rose to about 55% on the seventh day, and the trend was slowly rising. However, under the severe inflammation condition (Figure 8C), the PA encapsulation quickly failed in 4 days. The overall probability of failure increased rapidly, and all the samples failed completely on the fourth day. It is of note that, in the high H_2_O_2_ concentration environment, PA packaging can only guarantee a survival rate of high than 50% within 2 days. The situation of Si_3_N_4_ was similar to PA. Particularly, in the PBS solution, the trendline of Si_3_N_4_ in the first three days was similar to that of PA. Then the probability of failure continuously increased after day 3 and reached 60% on day 7. In the PBS solution with a lower concentration of H_2_O_2_, the trend in the first three days was still close to PA, and rapidly increased to 80% on day 7. However, in the high concentration H_2_O_2_ + PBS solution, the failure rate of the Si_3_N_4_ sample was lower than PA with a day-7 failure rate of 80%, indicating that the sample still had a small probability of survival (~20%). The case of SiO_2_ is very special, in which the SiO_2_ package was rapidly damaged in all the three testing environments under both the open-tip and closed-tip configurations. This indicates that SiO_2_ deposited by PECVD with a thickness of 1 µm may not be suitable for use as a packaging material. PECVD’s deposition quality and internal stress problem are usually the cause of this result.

Figure 9 shows the SEM images of the tungsten wires with various coatings, obtained at the endpoint of the 7-day accelerated test in the reactive solutions.

The pattern and severity of the metal damage were observed as a preliminary assessment of the potential weak points of the package. Due to the dissolution of H_2_O_2_ on the tungsten wire, different packaging failure mechanisms can cause different metal damage patterns. Without deliberate opening, the closed tip design makes the package relatively complete. The damage is mostly the “breakage” type as shown in Figure 9D. This is usually due to H_2_O_2_ diffusion through pinholes or microcracks in the middle section of the package, which causes the local chemical reaction with tungsten and therefore the wire breakage in the middle while the rest parts are generally intact. As shown in Figure 9A, although it is also a closed tip design, severe damage of “metal crack” still occurred, indicating that this segment of metal is completely exposed to H_2_O_2_ due to the delamination failure of the protective layer over this segment. This coincides with the poor performance of the SiO_2_ layer in impedance stability. The open-tip design deliberately leaves an opening that makes the dissolution effect more intense. This intensive dissolution effect had led to, as shown in Figure 9, the tested samples to appear“metal expansion” or “sharpening” damage. In the “tunnel” type (Figure 9E), the solution would corrode the interior metal wire along the opening, even though the exterior package remained relatively intact. Due to the relatively good performance of the PA package, no unique metal damage pattern was identified in the SEM images of the PA costed samples. In general, the damage patterns of the PA coated wires are similar to those of the Si_3_N_4_ coated wires.

Without H_2_O_2_, the PBS solution has a weak erosion effect on tungsten so that the failure mechanisms of different protective coatings can be more clearly observed. The closed-tip and open-tip designs did not show significant differences in the failure modes of the packaging materials. As shown in Figure 10C, PA was observed to have perforation failure, while Si_3_N_4_ suffered from both perforation and delamination (Figure 10A). For the SiO_2_ package, fragmentation and delamination failure were noticed as shown in Figure 10B,D, respectively.

### 3.3. Composite Packaging Design

Under the open-tip configuration, various composite coatings were prepared by PECVD growth of a 100 nm non-organic material (Si_3_N_4_ or SiO_2_) on tungsten wires as the bonding layer followed by 1 µm PA deposition. As shown in Figure 11A, in the PBS solution, the stability of the tungsten wire protected by the composite material was improved compared to the single material coating. Compared to the single material coating with Si_3_N_4_ or SiO_2_ only, the wire failure rate on day 1 was reduced significantly from 50% to 20% for the Si_3_N_4_+PA coating, and from 60% to 40% for the SiO_2_ + PA coating. The change in Si_3_N_4_ + PA was relatively significant, with the average failure rate reduction by 20% in the first four days. However, it must be pointed out that, in the inflammatory environment, the open-tip composite protective layer does not improve the survival rate of the tungsten wire effectively. Also, as can be seen from the comparison of a single material group, while the packaging performance of the composite materials was improved compared to a single inorganic layer, their protective performance was still worse than PA. Limited by the deposition temperature of PECVD (~300 °C) and the temperature endurance limit of PA (with a glass transition temperature of ~ 90 °C), PA as the bonding layer of the composite material is not studied.

The SEM images further verify that the composite strategy has a certain improvement in protective ability compared to a single inorganic material. As shown in Figure 12, in the PBS solution, a perforation phenomenon appeared on the Si_3_N_4_ + PA package, while SiO_2_+PA showed delamination failure. In the H_2_O_2_ environment (30 mM H_2_O_2_ + PBS), the tungsten wire showed “breakage” and “metal delamination” damage. From the comparison between Figure 9 and Figure 12, it can be found that, although the composite strategy still suffers from the sample failure caused by package breakage, failure mechanisms such as package cracking or metal sharpening were effectively reduced or not observed.

### 3.4. Multilayer Stack Strategy and Thickness Effect

From the above results, it can be concluded that PA has the most stable protective performance among the three materials. Then, it is necessary to further explore the potential of PA under different packaging strategies. In this case, tungsten wires were coated with three stacked PA layers (1 µm for each layer and 3 µm in total) and compared with wires coated with a single, 3 µm PA layer. The closed-tip design was adopted to slow down the erosion of tungsten. Figure 13 shows the experimental results where the stacked deposition does not show a significant difference from the single-layer one. In both the PBS and low concentration H_2_O_2_ (30 mMol H_2_O_2_ + PBS) environments, the performance of those two deposition strategies had slight differences in their failure rates over the testing period. However, on day 7, the difference between those two strategies was less than 10%. Moreover, under the high H_2_O_2_ concentration, the failure rate curves of the two coatings almost overlaid with each other, which further confirms that the stacked deposition does not necessarily improve the protective performance of PA.

The effect of PA thicknesses on packaging performance was also analyzed. As shown in Figure 14, a thicker PA coating effectively reduced the failure rate in the PBS environment.

In the environments containing H_2_O_2_, the 3 µm PA coating did not perform better than the 1 µm PA samples. In fact, the failure rates of the 3 µm samples were higher than those of the 1µm samples, and the average failure rate in seven days was 10% higher, as shown in Figure 14B,C. Considering the potential variation in PA deposition quality, the overall stability of the 3 µm PA coating is not significantly different from the 1 µm one. On the other hand, the 5 µm PA packaging, exhibited outstanding packaging stability throughout the RAA testing. No wire damage or packaging failure was detected in all the tested samples in 7 days. This result suggests that PA packaging of ≥ 5 µm has the potential to protect neural implants for ~56 days at the body temperature of 37 °C, estimated using Equation (4).

The results in Figure 15 show that the PA thickness has a limited impact on the packaging performance of PA in the open-tip configuration. In all three solutions, the failure rate curves of the 5 µm and 1 µm PA packages were almost identical. This could be attributed to the diffusion of moisture and ions into the PA-tungsten interface that weakens the bonding strength of PA, and therefore, accelerated the erosion of the tungsten wire. In the open-tip design, tungsten corrosion is believed to be the dominating failure factor.

## 4. Discussion

### 4.1. PA Coating and Thickness Effect

Table 2 shows all specifications and survival times of the PA coating samples, where 3-day and 7-day data represent the number of samples surviving for more than 3 days and 7 days, and the proportion relative to the total sample number. When the freshly prepared samples are immersed in the hot solution, some samples may fail due to manufacturing flaws and thermal mismatch [39,40]. The purpose of listing the 3-day and 7-day data is to eliminate such effects. A low 3-day survival rate indicates that the additional damages (e.g., thermal mismatch) may be high, or this packaging design is not suitable for the experimental environment. If the difference between the 3-day and 7-day data is not significant, it means that the failure of the sample at the beginning may be caused by manufacturing defects, and the 3 to 7 day data is closer to the real sample failure rate over time. The average survival time (AST) indicates the average survival time of failed samples, by which the estimated survival time of different packaging designs can be preliminary calculated and better compared. The average equivalent survival time (ASET) at 37 °C was calculated based on Equation (4) to estimate the possible average survival time of the sample in vivo. Under all the test conditions, PA exhibits the most stable packaging performance over the other two tested inorganic materials. In the closed-tip design, devices coated with thicker PA normally show better reliability than the ones coated with thinner PA. Comparing the PA samples of different thicknesses tested in the PBS solution, the survival rate of the 1 μm samples reaches 70% at day-3, while the 3-day survival rate of the 3 μm samples reaches 83%. The AST of the 3 μm samples is also much higher than that of the 1 μm samples. The data shows that failures of the 3 μm samples typically occur in the last 4 days, while failures of the 1 μm samples occur mainly in the first 3 days. All the 5 μm samples survived more than 7 days. For the samples tested in the PBS + H_2_O_2_ (30 mM) solution, the 3-day survival rate and the AST value of the 3 μm samples are slightly higher than those of the 1 μm samples. This indicates that thickness increase to 3 μm does not improve the packaging performance of PA, and the slight difference in the survival rates could be caused by manufacturing variance. When the PA thickness was increased to 5 μm, the 7-day survival rate of the samples was 100%, improved by 60% compared to the 1 μm samples. The performance improvement of the PA package does not depend linearly on the increase in PA thickness. Compared to the 3 μm, single PA deposition samples, the PA-S (3 μm stacked PA deposition) samples show a slightly higher 7-day survival rate in the PBS but a lower 7-day survival rate in PBS + H_2_O_2_ solutions. This could be caused by the failures at PA-to-PA interfaces, such as delamination, due to weak bonding strength between PA layers. However, for the open-tip design, the increase in PA thickness has no significant influence on reducing the failure probability. In such cases, the solution directly attacks the exposed metal, causing severe delamination of PA from the tungsten wire and eventually the complete failure of the package. This resulted in no significant difference between 1 μm and 5 μm in the 3-day and 7-day survival rates, and most samples were damaged on the first day (AST < 1). Under the same thickness, the stack deposition strategy does not show an overwhelming advantage compared to the single coating strategy, due to the relatively weak bonding strength between PA layers in reactive solutions. As a result, delamination may occur between layers which will greatly affect the reliability of the stacked coating. Considering the above reasons, further improvement in PA bonding quality will be an effective method to enhance the reliability of multiple stacked packages.

### 4.2. Non-Organic Coating and PECVD

While Si_3_N_4_ and SiO_2_ are very stable and chemically resistant, the built-in stress in Si_3_N_4_ and SiO_2_ after PECVD significantly affected their packaging performance. The stress can cause cracks in the thin film, resulting in device failure after a short period. As shown in Table 3, most of the open-tip samples cannot survive for more than 1 day in the environments containing H_2_O_2_, regardless of the packaging materials. For the close-tip samples, the 7-day survival rate of the Si_3_N_4_ packaged samples reached 20% in PBS + H_2_O_2_ (150 mM) solution and the AST is about 4, which is better than the performance of thin PA package (1 μm and 5 μm). These data indicate that Si_3_N_4_ has great potential to act as a packing material for biomedical implants. Due to different thermal expansion coefficients of different materials, composite coatings tend to possess increased thermally-induced stress, which accelerates the failure of the packages. These problems led to the relatively poor reliability of non-organic materials in our experiments. In terms of composite coating, while the current results are too preliminary due to limitations in materials and packaging equipment, there are still two points worth pointing out. First, although not obvious, composite materials do improve stability compared to single inorganic materials (Figure 11). In Table 3, the AST of the composite design is slightly better than that of the non-organic single coating in PBS solution, indicating that the composite design can extend the survival time to a certain extent.

Their failure rates are slightly lower than single-layer inorganic materials, and higher than the single-layer PA package. Second, the bonding layer in the two-layer simple superimposed structure has the greatest effect on the packaging result. In the fabrication of composite packaging with the superimposed structure, the intermediate layer should form strong bonding with both the metal and the subsequent polymer coating in order to achieve the best packaging performance.

### 4.3. Evaluation of Failure

It is worth noting that the purpose of using the open-tip probe is to simulate a real device. In this study, we found that the majority failure mode is delamination and then metal erosion that progresses along the interface between the tungsten wire and the packaging layer. As such, the devices failed even if the outer package remained relatively intact. Enhancing the bonding strength between inhomogeneous material coatings could be a solution to delamination, thereby making the package more reliable and effective. An in-depth analysis of the failure data in Figure 8 suggests that the impedance of the tungsten wire abnormally increased before the wire broke. This can be caused by the formation of small cracks and defects within the package, resulting in localized tungsten corrosion that gradually increases the impedance. Finally, the concentration of H_2_O_2_ directly affects the stability of the package. The concentration of H_2_O_2_ is related to the severity of inflammation of the experiment subject. Therefore, besides improving the performance of the package, care should be taken to reduce the inflammatory response at the device-tissue interface to extend the service life of the implanted devices.

## 5. Conclusions

In this paper, we simulated the environments of cortical neural implantation with different degrees of inflammations by adding distinctive concentrations of the reactive chemical. We used those inflammation scenarios to investigate the effects of accelerated aging of tungsten wires packaged using different strategies. This RAA approach allows us to quickly evaluate different packaging strategies and packaging materials to find the best packaging solution, and make the results close to real in vivo data. While preliminary, our results show that PA is the most stable material among the three tested materials (PA, SiO_2_, Si_3_N_4_) and under the same test conditions. Thicker PA films exhibit better reliability over time. Second, significant delamination and fragmentation occurred in PECVD SiO_2_ films after a short period of soaking, resulted from thermally induced stress due to the mismatch of the thermal expansion coefficients between SiO_2_ and tungsten. Third, the composite coatings combining organic and inorganic films had slightly improved packaging performance compared to single inorganic material encapsulation. Lastly, the failure of the open-tip wires was mostly caused by delamination at the metal-package interface regardless of the integrity of the packaging films. Techniques to enhance the interface bonding strength is critically needed for further improvement of the packaging performance of these materials. In the future, based on the presented testing procedures and data analysis method, more materials will be evaluated, such as the SiC, polyimide, and diamond-like carbon. As the scope of materials continues to expand, a comprehensive database will be established for packaging materials used in biomedical implants.

## Figures and Tables

**Figure 1 micromachines-11-00810-f001:**
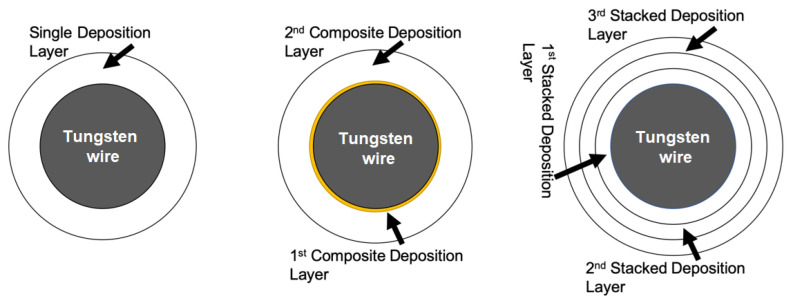
Schematic diagram of the three wire packaging categories.

**Figure 2 micromachines-11-00810-f002:**
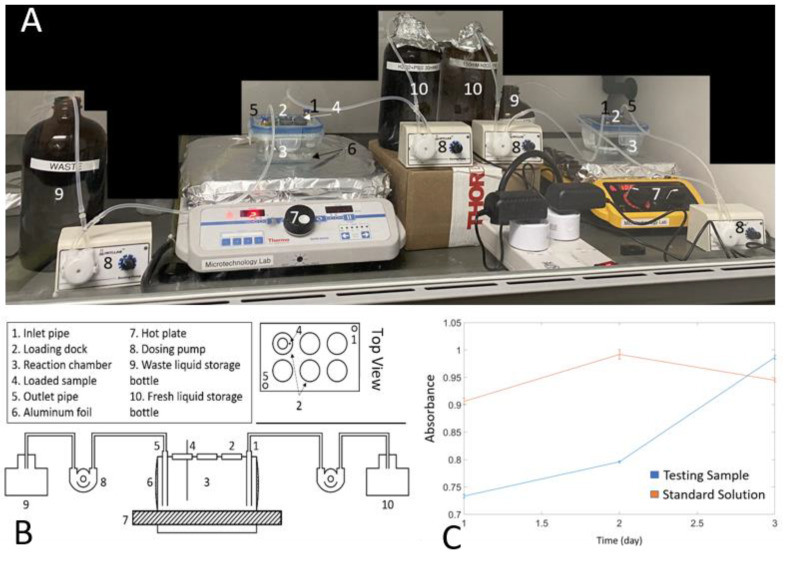
(**A**) RAA test platform, (**B**) Schematic diagram of the flow cell, (**C**) UV-visible spectroscopy of 30 mM H_2_O_2_ + PBS testing solution comparing with standard solution (freshly prepared H_2_O_2_ + PBS (30 mM) solution).

**Figure 3 micromachines-11-00810-f003:**
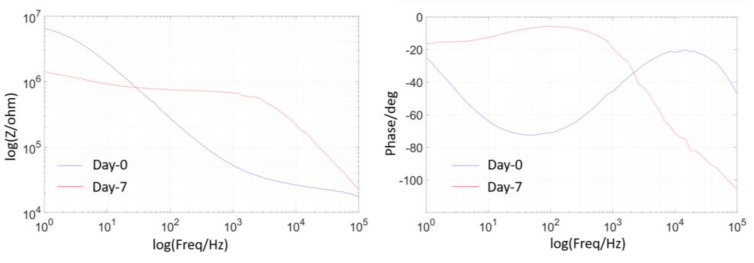
EIS plots of a 1 μm PA coated, open tip sample soaked in PBS + 30 mM H_2_O_2_ solution at Day-0 to Day-7.

**Figure 4 micromachines-11-00810-f004:**
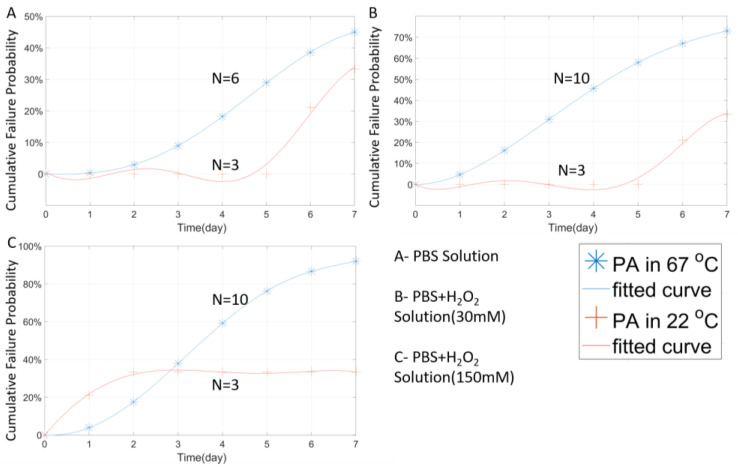
The Weibull cumulative distribution data and corresponding curve fitting of the closed-tip samples coated with 3 µm stacked-deposition of PA, measured at two temperatures of 67 °C or 22 °C in different solutions.

**Figure 5 micromachines-11-00810-f005:**
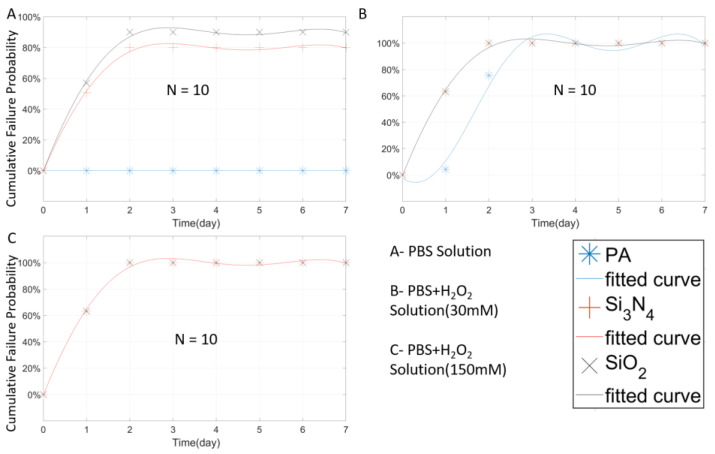
The Weibull cumulative distribution data and corresponding curve fitting of open-tip tungsten wires with a single layer coating (1 µm) of PA, Si_3_N_4,_ and SiO_2_, tested at 67 °C in different solutions (the curves in B and C may overlap).

**Figure 6 micromachines-11-00810-f006:**
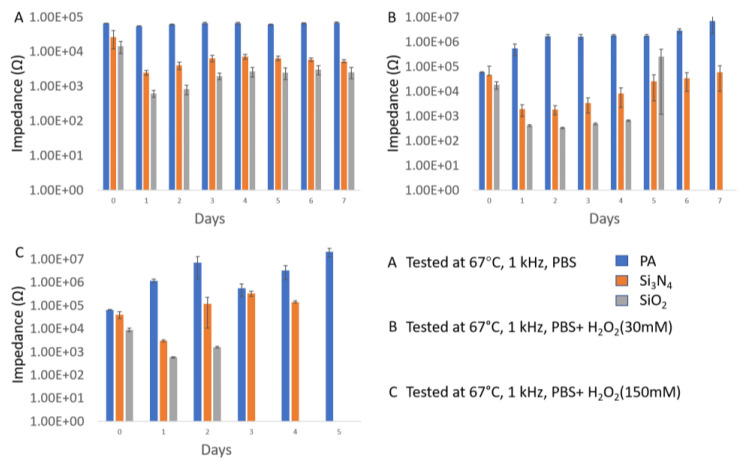
Impedance changes of open-tip tungsten wires coated with a single layer (1 µm) of PA, Si_3_N_4_, and SiO_2_, tested at 67 °C, 1 kHz, in three different solutions.

**Figure 7 micromachines-11-00810-f007:**
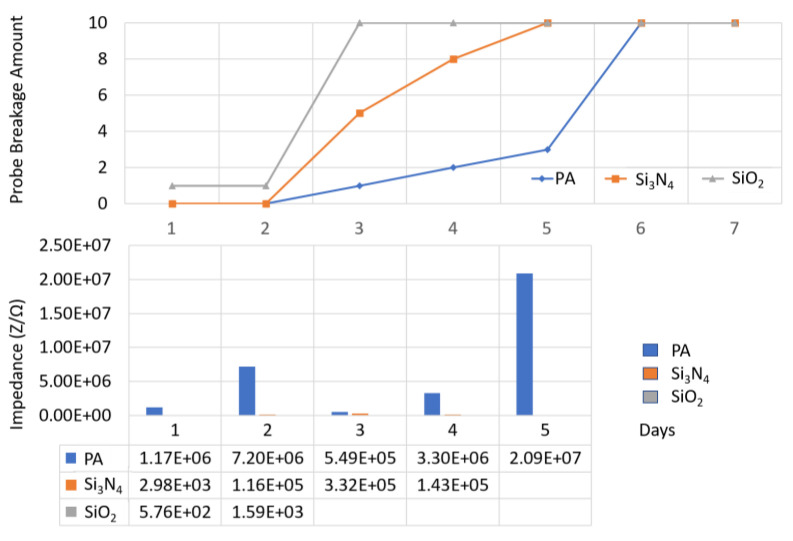
Average impedance change and wire breakage for samples with a single layer coating (1 µm) of PA, Si_3_N_4_, and SiO_2_, tested at 67 °C, 1 kHz, PBS + H_2_O_2_ (150 mM).

**Figure 8 micromachines-11-00810-f008:**
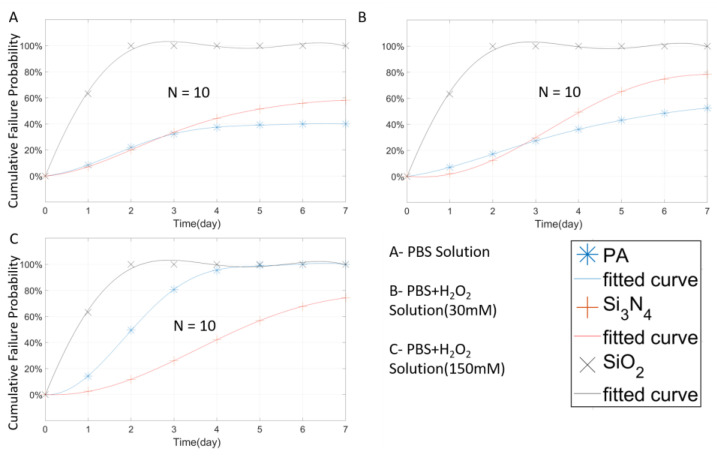
The Weibull cumulative distribution data and corresponding curve fitting of the closed-tip tungsten wires with a single layer coating (1 µm) of PA, Si_3_ N_4_, and SiO_2_, tested at 67 °C in different solutions.

**Figure 9 micromachines-11-00810-f009:**
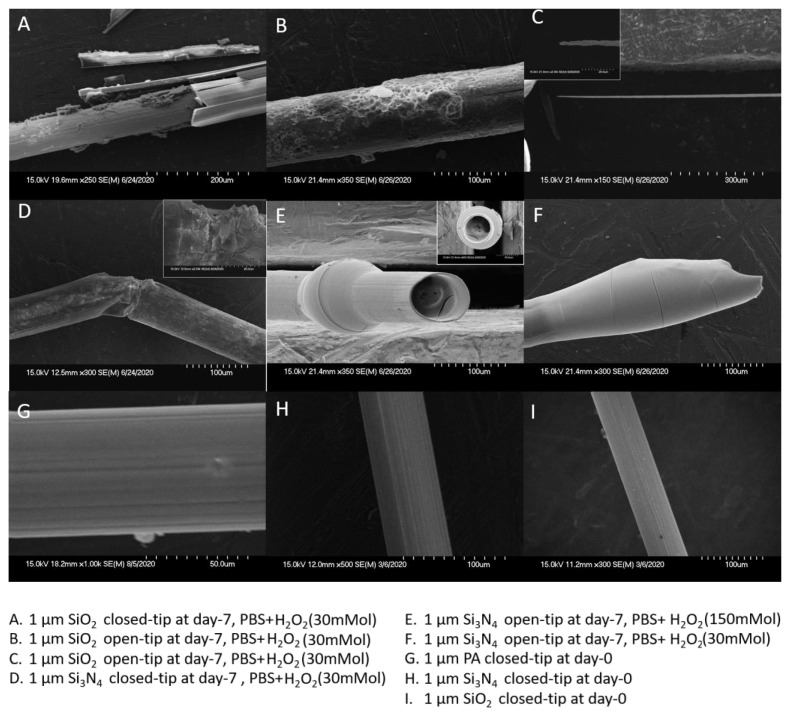
Six (6) damage patterns of the tungsten wires observed in H_2_O_2_ + PBS solution: (**A**) Metal crack, (**B**) Perforation, (**C**) Sharpening, (**D**) Breakage, (**E**) Tunnel, (**F**) Metal expansion, and (**G**–**I**) were the SEM images of untested samples.

**Figure 10 micromachines-11-00810-f010:**
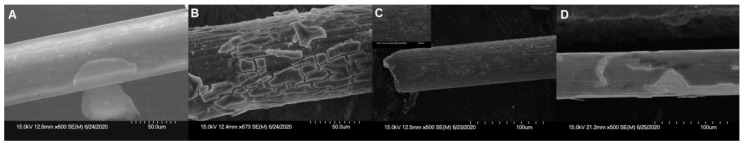
Three (3) failure modes of the package materials observed in PBS solution, (**A**) Package delamination on 1 μm Si_3_N_4_ closed-tip, (**B**) Package Fragmentation on 1 μm SiO_2_ closed-tip, (**C**) Package Perforation on 1 μm PA closed-tip, (**D**) Package delamination on 1 μm SiO_2_ open-tip.

**Figure 11 micromachines-11-00810-f011:**
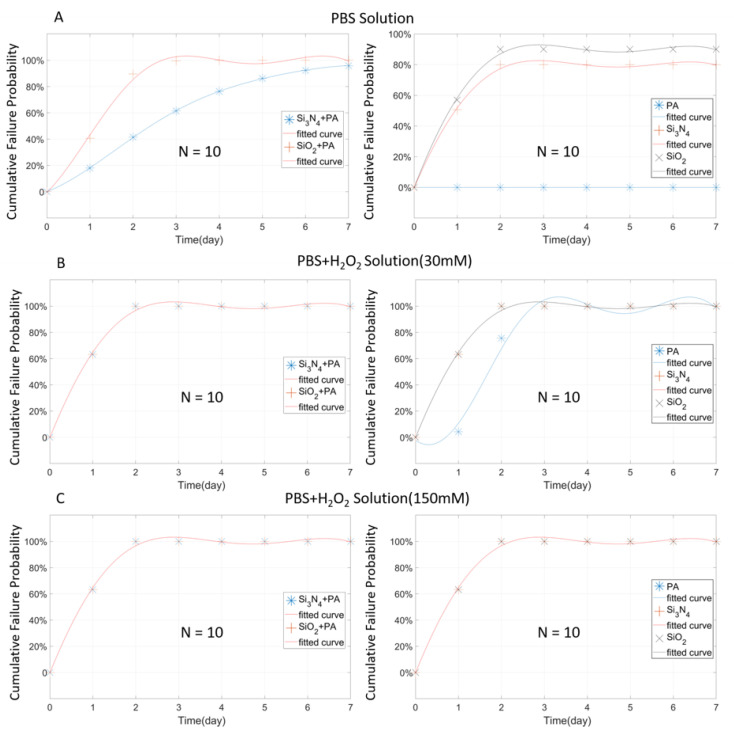
The Weibull cumulative distribution data and corresponding curve fitting of the open-tip tungsten wires coated with combined films of Si_3_N_4_ (100 nm)/PA(1 µm) and SiO_2_ (100 nm)/PA(1 µm) (**left side**) versus the Weibull cumulative distribution data and corresponding curve fitting of open-tip wires with a single layer coating (1 µm) of PA, Si_3_N_4_ and SiO_2_ (**right side**), tested at 67 °C in different solutions.

**Figure 12 micromachines-11-00810-f012:**
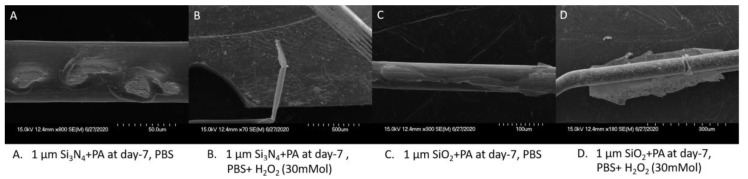
Metal damage observed in H_2_O_2_ + PBS and PBS solutions.

**Figure 13 micromachines-11-00810-f013:**
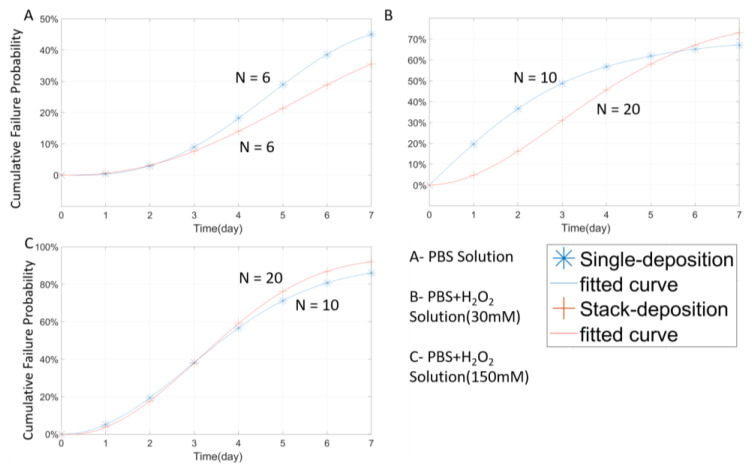
The Weibull cumulative distribution data and corresponding curve fitting of the closed-tip samples with stacked-deposition (1 µm + 1 µm + 1 µm) and single-deposition (3 µm) of PA under the same thickness. Experiments were performed in different solutions at 67 °C.

**Figure 14 micromachines-11-00810-f014:**
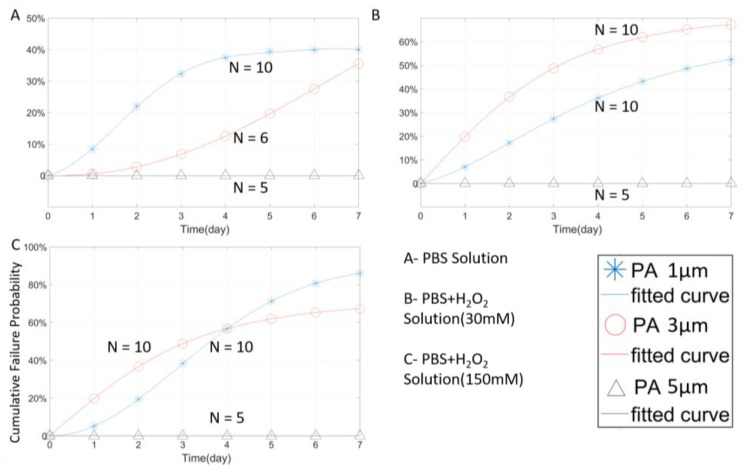
The Weibull cumulative distribution data and the corresponding curve fitting of closed-tip samples with 1 µm, 3 µm, and 5 µm PA. Experiments were performed in different solutions at a temperature of 67 °C.

**Figure 15 micromachines-11-00810-f015:**
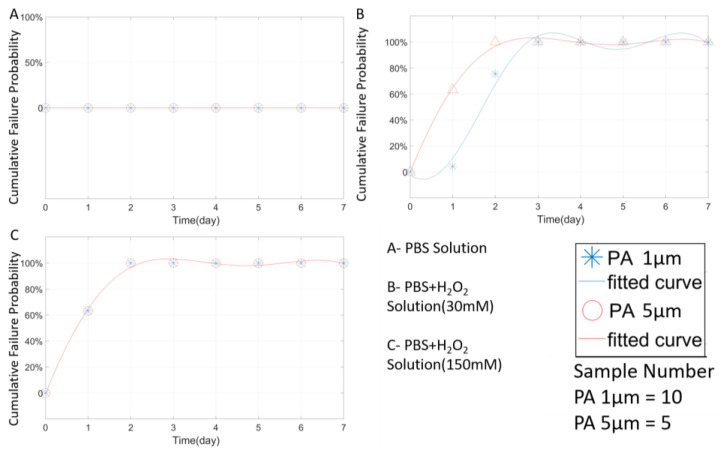
The Weibull cumulative distribution data and corresponding curve fitting of open-tip samples with 1 µm and 5 µm PA coating. Experiments were performed in different solutions at a temperature of 67 °C.

**Table 1 micromachines-11-00810-t001:** All tested sample and their configurations in three solutions.

Material	Thickness (μm)	Type	Samples Number	Time (Days)	Temperature (°C)
SiO_2_	1	Open-tip	30	7	67
Si_3_N_4_	1	Open-tip	30	7	67
PA	1	Open-tip	30	7	67
SiO_2_	1	Close-tip	30	7	67
Si_3_N_4_	1	Close-tip	30	7	67
PA	1	Close-tip	30	7	67
SiO_2_ + PA	1.1	Open-tip	30	7	67
Si_3_N_4_ + PA	1.1	Open-tip	30	7	67
PA	3	Close-tip	26	7	67
PA	3	Close-tip	9	7	22
PA-S ^1^	3	Close-tip	46	7	67
PA	5	Close-tip	15	7	67
PA	5	Open-tip	15	7	67

^1^ PA-S: stacked deposition PA samples.

**Table 2 micromachines-11-00810-t002:** Survival probabilities of all PA coating samples.

Material	Thickness (μm)	Type	Solution	3-day ^4^	7-day ^5^	AST ^6^ (Day)	AEST ^7^ (Day)	LST ^8^ (Day)	LEST ^9^ (Day)
PA	1	Open-tip	PBS	10(100%)	10(100%)	NA	NA	NA	NA
PA	5	Open-tip	PBS	5(100%)	5(100%)	NA	NA	NA	NA
PA	1	Open-tip	P + H(30) ^2^	0(0%)	0(0%)	<1.7	<13.6	<2	<16
PA	5	Open-tip	P + H(30)	0(0%)	0(0%)	<1	<8	<1	<8
PA	1	Open-tip	P + H(150) ^3^	0(0%)	0(0%)	<1	<8	<1	<8
PA	5	Open-tip	P + H(150)	0(0%)	0(0%)	<1	<8	<1	<8
PA	1	Close-tip	PBS	7(70%)	6(60%)	<2	<16	<4	<32
PA	3	Close-tip	PBS	5(83%)	3(50%)	<4.6666	<37.333	<6	<48
PA-S ^1^	3	Close-tip	PBS	5(83%)	4(66%)	<4.5	<36	<2	<16
PA	5	Close-tip	PBS	5(100%)	5(100%)	NA	NA	NA	NA
PA	1	Close-tip	P + H(30)	7(70%)	3(30%)	<3.833	<30.6667	<7	<56
PA	3	Close-tip	P + H(30)	5(50%)	3(30%)	<2.42	<19.42	<6	<48
PA-S	3	Close-tip	P + H(30)	12(60%)	4(20%)	3.875	31	<6	<48
PA	5	Close-tip	P + H(30)	5(100%)	5(100%)	NA	NA	NA	NA
PA	1	Close-tip	P + H(150)	2(20%)	0(0%)	<2.1	<16.8	<4	<32
PA	3	Close-tip	P + H(150)	6(60%)	1(10%)	<3.5555	<28.444	<6	<48
PA-S	3	Close-tip	P + H(150)	10(50%)	1(5%)	3.57	28	<6	<48
PA	5	Close-tip	P + H(150)	5(100%)	5(100%)	NA	NA	NA	NA

^1^ PA-S: Stacked deposition PA samples; ^2^ P+H(30): PBS + H_2_O_2_ (30 mM) solution; ^3^ P+H(150): PBS + H_2_O_2_ (150 mM) solution; ^4^ 3-day: 3 day survival sample number (% of total); ^5^ 7-day: 7 day survival sample number (% of total); ^6^ AST: Average Survival Time of Failed Samples; ^7^ AEST: Average equivalent (37 °C) Survival Time of Failed Samples, calculated by Equation (4); ^8^ LST: Longest Survival Time of Failed Samples; ^9^ LEST: Longest Equivalent (37 °C) Survival Time of Failed Samples, calculated by Equation (4).

**Table 3 micromachines-11-00810-t003:** Survival probabilities of all non-organic coating and composite design samples.

Material	Thickness (μm)	Type	Solution	3-day	7-day	AST (Day)	AEST (Day)	LST (Day)	LEST (Day)
SiO_2_	1	Open-tip	PBS	1(10%)	0(0%)	<1.3	<10.4	<4	<32
Si_3_N_4_	1	Open-tip	PBS	2(20%)	2(20%)	<1	<8	<1	<8
PA	1	Open-tip	PBS	10(100%)	10(100%)	NA	NA	NA	NA
SiO_2_	1	Open-tip	P + H (30)	0(0%)	0(0%)	<1	<8	<1	<8
Si_3_N_4_	1	Open-tip	P + H (30)	0(0%)	0(0%)	<1	<8	<1	<8
PA	1	Open-tip	P + H (30)	0(0%)	0(0%)	<1.7	<13.6	<2	<16
SiO_2_	1	Open-tip	P + H(150)	0(0%)	0(0%)	<1	<8	<1	<8
Si_3_N_4_	1	Open-tip	P + H(150)	0(0%)	0(0%)	<1	<8	<1	<8
PA	1	Open-tip	P + H(150)	0(0%)	0(0%)	<1	<8	<1	<8
SiO_2_	1	Close-tip	PBS	0(0%)	0(0%)	<1	<8	<1	<8
Si_3_N_4_	1	Close-tip	PBS	7(70%)	4(40%)	<3	<24	<6	<48
PA	1	Close-tip	PBS	7(70%)	6(60%)	<2	<16	<4	<32
SiO_2_	1	Close-tip	P + H (30)	0(0%)	0(0%)	<1	<8	<1	<8
Si_3_N_4_	1	Close-tip	P + H (30)	8(80%)	2(20%)	<3.625	<29	<5	>40
PA	1	Close-tip	P + H (30)	7(70%)	3(30%)	<3.833	<30.6667	<7	<56
SiO_2_	1	Close-tip	P + H(150)	0(0%)	0(0%)	<1	<8	<1	<8
Si_3_N_4_	1	Close-tip	P + H(150)	6(60%)	2(20%)	<4	<32	<7	<56
PA	1	Close-tip	P + H(150)	2(20%)	0(0%)	<2.1	<16.8	<4	<32
SiO_2_ + PA	1.1	Open-tip	PBS	0(0%)	0(0%)	<1.2	<9.6	<3	<24
Si_3_N_4_ + PA	1.1	Open-tip	PBS	4(20%)	0(0%)	2.8	22.4	<7	<56
SiO_2_ + PA	1.1	Open-tip	P + H (30)	0(0%)	0(0%)	<1	<8	<1	<8
Si_3_N_4_ + PA	1.1	Open-tip	P + H (30)	0(0%)	0(0%)	<1	<8	<1	<8
SiO_2_ + PA	1.1	Open-tip	P + H(150)	0(0%)	0(0%)	<1	<8	<1	<8
Si_3_N_4_ + PA	1.1	Open-tip	P + H(150)	0(0%)	0(0%)	<1	<8	<1	<8

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
