# Peer review of "Stability Performance Analysis of Various Packaging Materials and Coating Strategies for Chronic Neural Implants under Accelerated, Reactive Aging Tests"

_micromachines, 2020, doi:10.3390/mi11090810_

Round 1
Reviewer 1 Report
The manuscript micromachines-869334 'Stability Performance Analysis of Various Packaging Materials and Coating Strategies for Chronic Neural Implants under Accelerated, Reactive Aging Tests' by Gong and co-workers describes a systematic study for different passivation materials of cortical neuronal implants. The authors used accelerated aging techniques and utilized impedance spectroscopy and optical- as well as electron microscopy to characterize material failures.
The results of this study are interpreted very carefully and self-reflecting. To my opinon, the experimental data are very significant! The manuscript is well-structured and the data are displayed in a clear and straightforward way.
The methods are described clearly as well, so that other groups could reproduce the results. This paper will attract a wide readership since passivation materials of neural electrodes are a well-known problem. I also appreciate that “negative” results are presented here as well!
Before consideration of acceptance, I do have following minor comments to improve the quality of this manuscript (lines):
29: Reference is missing! RAA with H2O2 is a well-established procedure for long-term stability testing in neural implants
42: “Two device configures configurations were investigated:…”
52: Why 50% of initial impedance? Isn’t e.g. 75% also severe damage to the electrodes?
80-98: In general it would be nice to have one table which contains all the experiments regarding materials, solutions, amount of electrodes, compositions etc…
87: “Second, a composite deposition design is used to study the interaction of the two materials”
112: Fig. 2: Maybe use a new picture without the black paint
136: “… using closed-tip 3um µm PA coated tungsten wires” not consistent, all the other times µm is used
143: Figure 3: What is the standard solution? Pure PBS?
169: Where are the curves for day 7? (what is the difference between the purple and the blue curve?)
207: Figure 5: What are probabilities <0 ? (Also if this is due to the fitting parameters a short explanation here would be nice)
233: Figure 6: Why are not all three different materials tested in all three solutions?
290-294: I don’t understand this explanation. Of course, not every PA deposition is the same and there could be some unevenness. In how many different runs were the tungsten wires coated? I would expect that in average the layers should be comparable to those from the open-tip deposition run. For the open-tip wires in PBS there was almost no change in the impedance measurements in 7 days. So the explanation, that in open-tip wires the exposed area is largest and this dominates the impedance changes does not make sense for PBS and is also not explaining the curve in figure 9A.
329: Sentence could be also understood that it is shown that a lot of wires showed “metal expansion” and “sharpening”. Maybe use a different formulation here.
330: “…, it shows a very unique damage patterns”.
378: Figure 14: Why has B such a different shape than A and C ?
407: This is a very interesting information. It would be nice to have this “realtime projection”, so the calculations with eq. 4 more often!
477: “Third, The the…”
Based on my above comments I recommend a minor revision of this manuscript.
Author Response
We thank the reviewer's valuable comments, please see the attachment.

Reviewer 2 Report
In this manuscript, the authors studied the stability of Parylene C (PA), SiO2, and Si3N4 packages and coating strategies on tungsten wires using accelerated, reactive aging tests, since the reliable packaging for implantable devices in body fluids is a long-standing challenge for device’s chronic applications. On the whole, it could be a meaningful work that can provide some helpful information to researchers interested in implantable devices based biointerfaces. However, there are some logic and experimental design problems in this work. Therefore, the reviewer suggests that major revision of this manuscript is required.
- . The control group in this work was set at a room temperature (22 oC), so why did not use 37 oC to simulate the physiological state?
- . In practical situations, the PH value always varies with hydrogen peroxide concentration in body fluids. The changed PH will also affect the package performance severely. By considering this, the research will make more sense.
- . For the experiments with varied temperature, only closed tip wires were used. The failure mechanism of open tip wires may also be different under varied temperature. Thus, this part of experiments should be considered.
- . About the figures, firstly, there are 16 figures in total. It is better to integrate some of them together. Secondly, the formats of the figures should be optimized uniformly. Thirdly, the label statements and the chemical formula in figure 4 and figure 7 are incorrect respectively.
- . In order to observe morphology changes of tungsten wire samples, only the tested samples’ SEM images are provided in this manuscript. To have better understanding and comparison, the pristine SEM images of untested samples are also needed.
- . Line 289 Page 10 mentions that the PA coated closed-tip wires show poor stability than the open-tip ones. That may due to the different impedance composition of these two kinds of wires. Thus the impedance changes of closed-tip wires may not means the failure.
- . In the discussion part, Line 421-436 Page 17, the author tried to discuss the PA coating thickness effect on anti-corrosion. However, this discussion did not make much sense because the authors did not give any quantitative or detailed value analysis. The same applies to the conclusion part. Both quantitative analysis and systematic conclusion are very important for a scientificresearch.
Author Response

(The authors gave the same response as above.)

Reviewer 3 Report
The authors present results on the effect of RAA on dielectric coatings on tungsten microwires to be used in neural implants.
Scope: One of my concerns is that vapor deposition of dielectric coatings on tungsten microelectrodes is not used in neural engineering applications. Polyimide-encapsulated tungsten wires are the benchmark implantable wire and should have been discussed and/or compared in the study. If tungsten was only used to easily show film degradation because of its corrosive nature in PBS +H2O2, then it needs to be more clearly defined. The use of "packaging materials" in neural implants are different than electrode dielectric materials. There needs to be a more clear definition of what you mean by the material choice in your study.
Much work has been done by others on thin-film dielectric encapsulation materials for planar neural interfaces and they are not referenced in this work. Many use a hybrid combination of deposited materials for robust sticking and coverage. SiC, for example, is a current material of interest. Thus, it is hard to ascertain if any of the presented work is novel. The introduction and need for the study needs to be much improved.
Methods: I do not think that the metric of % reduction in impedance at 1 kHz is a good metric for failure. Impedance is greatly dependent on the metal surface area exposed and thus there will be different relationships for the "open tip" and the "closed tip" cases. For example, Figure 4 (even thought he data legend is inaccurate) shows that the impedance at 1 kHz increases after 7 days in solution. Yet, the metric for the failure was impedance decrease of 50%.
-
190 The MTTF was determined when the measured 1 kHz impedance of the samples was below
-
191 50% of the initial value.
The impedance data presented in Figure 7 shows variable decrease and increase in impedance over time. Since impedance is greatly affected by the surface area, evaluation at one value has to be done under very stringent conditions. See this paper for detailed use of impedance data over time.
Corrosion of tungsten microelectrodes used in neural recording applications E Patrick, ME Orazem, JC Sanchez, T Nishida - Journal of neuroscience methods, 2011
Impedance decrease at 1k over time could be due to film cracking, etc. since it would increase the surface area of the exposed metal. This could be seen on closed tip devices. However, with the open electrode with 3 mm of surface exposed, as the tungsten dissolves away, the surface area decreases thus the impedance will increase as shown in Figure 4.
In summary, the metrics for failure must be more tightly regulated for any consistency in the reported results to be believed. Plus the authors need to do a better job of referencing and reviewing prior literature on thin film materials in neural implants and the use of impedance data for analysis.
Author Response

(The authors gave the same response as above.)

Round 2
Reviewer 2 Report
The revised manuscript addressed the reviewer's comments accordingly. I suggest the paper be accepted.
Author Response
We are grateful for the valuable comments of the reviewer, and we have learned a lot during the process of revising.
Reviewer 3 Report
0. The authors still do not do a good job of referring to previous work. There are many inaccuracies.The introduction does not adequately describe the need for this work. And the new addition at the end of the intro is inaccurate.
"The main goal of this research is to build a test framework, by introducing 61the RAA test method to accelerate and simulate different degrees of inflammation in the body, and 62combining with the Weibull probability model to evaluate the risk of device failure. "
RAA is not introduced for the first time in this work, as the authors previously refer to Takamov et al. The Weibull probability model is new, but it needs to be better validated as a method. Correlation of impedance measurement and optical characterization for each probe is needed.
00. Throughout the methods and discussion section, previous work needs to be cited. I'm sure there have been many studies on the stability of PA, SiO2 and SiN thin films for microelectrode devices (as many commercial devices use PA, or SiO2). How does your study compare to theirs?
1. Major concern: The open-tip impedance assessment does not directly relate to package reliability. As the corrected text explains, the increase in impedance is always due to the dissolution of the metal surface. Thus it does not report any changes to the packaging layer. This study should be removed from the paper.
2. The use of 30 mM and 150 mM concentrations of H2O2 was not supported by citations. In fact, Takamov et al., say that 30 mM and above greatly overestimates the degree of inflammation seen in vivo.
"In this paper, we simulated the environments of cortical neural implantation with different degrees of inflammations by adding distinctive concentrations of the reactive chemical."
3. No description of the deposition parameters were given. Won't deposition parameters have an effect on the quality of the deposition? The authors need to comment how the deposition parameters were chosen and how they compare to other work. Also how was 360 degree coverage of the wire obtained in the deposition chambers? Where they suspended somehow?
4. I am not convinced that Tungsten is a good material to use as unbiased indicator of package integrity because it undergoes corrosion. The change in impedance will always be a combination of package failure and a change in impedance due to a changing active area. Why not just use Pt (a more widely used neural electrode material that does not corrode)? The use of Tungsten seems to muddy the package reliability results.
Also, Tungsten is only used with polyimide insulation in neural implants, mainly because commercial probes or wires are available. Innovation for the package material for Tungsten is not necessarily needed in the community. The following sentence in the text does not correctly cite [39]. That paper said that even though tungsten wires are readily used in the community, they should not be used for long term applications because of corrosion.
"Due to 75the strength, rigidity, and recording capability of tungsten, 50 μm diameter microwire becomes an 76excellent choice for intracortical applications[39]."
5. Takamov et al. say to beware of using only 1 kHz to characterize the probe... To make the failure probability data more believable, full spectra impedance plots should be given for the closed-tip cases. And an general trend of impedance change over time should be established for different failure modes (ie. pin holes, cracking, delamination etc.) and shown in the paper before the probability method can be trusted. For example, I would really like to know the reason for impedance fluctuation in Fig 6C.
6. I really think that the open tip data needs to be thrown out; it is not characterizing insulation package integrity...
&. My advice is to focus the paper on validating the Weibull characterization method by using Pt wires with closed-tip configurations. Then you will be comparing the quality of the insulation layer only. Back up the analysis with well documented full spectra impedance for each sample (show that impedance at 1 kHz is a stable and ok representation for failure) and corresponding failure mechanisms.
Author Response
We are grateful for the valuable comments of the reviewer, and we have learned a lot during the process of revising. Please see the attachment.
